# ARGs Detection in *Listeria Monocytogenes* Strains Isolated from the Atlantic Salmon (*Salmo salar*) Food Industry: A Retrospective Study

**DOI:** 10.3390/microorganisms11061509

**Published:** 2023-06-06

**Authors:** Gianluigi Ferri, Carlotta Lauteri, Anna Rita Festino, Alberto Vergara

**Affiliations:** Department of Veterinary Medicine, Post-Graduate Specialization School in Food Inspection “G. Tiecco”, University of Teramo, Strada Provinciale 18, Piano d’Accio, 64100 Teramo, Italy; gferri@unite.it (G.F.); clauteri@unite.it (C.L.); arfestino@unite.it (A.R.F.)

**Keywords:** *Listeria monocytogenes*, salmon, antibiotic resistance, resistance genes, molecular biology, food safety, One Health

## Abstract

Among bacterial foodborne pathogens, *Listeria monocytogenes* represents one of the most important public health concerns in seafood industries. This study was designed as a retrospective study which aimed to investigate the trend of antibiotic resistance genes (ARGs) circulation in *L. monocytogenes* isolates identified (in the last 15 years) from Atlantic salmon (*Salmo salar*) fresh and smoked fillets and environmental samples. For these purposes, biomolecular assays were performed on 120 *L. monocytogenes* strains collected in certain years and compared to the contemporary scientific literature. A total of 52.50% (95% CI: 43.57–61.43%) of these samples were resistant to at least one antibiotic class, and 20.83% (95% CI: 13.57–28.09%) were classified as multidrug resistant. Concerning ARGs circulation, tetracycline (*tet*C, *tet*D, *tet*K, *tet*L, *tet*S), aminoglycoside (*aad*A, *str*A, *aac*C2, *aph*A1, *aph*A2), macrolide (*cml*A1, *cat*I, *cat*II), and oxazolidinone (*cfr*, *optr*A, *poxt*A) gene determinants were majorly amplified. This study highlights the consistent ARGs circulation from fresh and processed finfish products and environmental samples, discovering resistance to the so-called *critical important antimicrobials* (CIA) since 2007. The obtained ARGs circulation data highlight the consistent increase in their diffusion when compared to similar contemporary investigations. This scenario emerges as the result of decades of improper antimicrobial administration in human and veterinary medicine.

## 1. Introduction

*Listeria monocytogenes* is one of the most important bacterial foodborne zoonotic pathogens representing a critical One Health concern. However, we must considerer that cases of infections and related outbreaks have decreased in the last decade [1]. Every year, this pathogen causes restricted numbers of confirmed hospitalization in the European and North American continents, after the ingestion of contaminated foodstuffs [2,3]. Consistent sanitary interests are especially focused on the effects on human pregnancy [1]. For this reason, European legislation included this microorganism in the so-called food safety criteria, as reported in the EU Reg. No. 2073/2005 [4]. Food production industries, and more specifically fish ones, have improved preventive measures in their microbiological risk assessment sections [5]. Periodic qualitative and quantitative microbiological screenings include stable parameters investigated in the surveillance programs monitored by the EFSA (Europe) and FDA (North America) [2].

Due to the environmental persistence of *L. monocytogenes* (relating to the optimal multiplication temperature in the range from 4 °C to 30 °C) and its ability to produce biofilm, fish food matrices represent ideal biochemical substrates for its microbial proliferation [6].

Primary, processed, and ready-to-eat fish products have been largely described in the scientific literature as potential *L. monocytogenes* drivers [7,8]. Among finfish species majorly traded in productive industrial systems, *Salmo salar* has been widely commercialized as a fresh or processed food matrix. Indeed, due to the high consumer demands for fish proteins, salmon aquaculture sector coupled with fish catching has provided the required tons [9]. However, the consequential high animal densities, both in mariculture (i.e., Scotland, Norway, etc.) and inland farming systems, have presented a growing need for antibiotic administration for therapeutic purposes to control infectious outbreaks in fish from hatching to adult age [10]. The massive or improper usage of these substances have produced selective pressures which have advantaged resistant bacterial strains. These acquired or intrinsic resistances can be driven by pathogenic and commensal bacteria and vice versa [11]. The so-called horizontal transmission of oligonucleotide determinants was identified as one of the evolutionary mechanisms to preserve specific bacterial strains. Indeed, it has relevant repercussions on animal and human health through the selection of multi- or pan-resistant isolates. Scientists have largely demonstrated that the cyto-biochemical interactions between consumers and foods microbiomes can be considered crucial sources for antibiotic resistance genes (ARGs) exchanges [12].

The expanding resistance patterns to the most used and legally allowed molecules in veterinary medicine (especially in the aquaculture zootechnic sector, as aminoglycosides, beta-lactams, quinolones, and tetracycline antimicrobial classes) have induced the introduction of alternative farming systems which have implemented preventive medicine measures, i.e., vaccination, probiotics, etc. [13]. Furthermore, from a comparative perspective, the increasing resistances to the CIA, the usage of which is indicated for humans only, have highlighted the notable importance of the antimicrobial resistance (AMR) phenomenon [14].

From fish products, many dated and recent studies have highlighted relevant resistance patterns against different antibiotic classes such as tetracyclines [15,16,17], aminoglycosides [16,18], and beta-lactams, which were recently discovered [19,20], and among CIA, bacteria were resistant to sulfonamides and carbapenems [21,22].

This retrospective study aims to perform a biomolecular investigation regarding the genetic determinants expressed as ARGs in 120 *L. monocytogenes* strains isolated from fresh and smoked Atlantic salmon (*Salmo salar*) fillets and environmental samples (i.e., surfaces, knives, and machines) in the fish industry. Therefore, this study was furtherly designed to produce original data, investigating gene determinants distribution and evolution in the last 15 years. The obtained retrospective results were compared with other 2008-aged studies and to the most recent ones from bibliographic data. This research, based on a wide application of biomolecular assays, reviews historical changes of ARGs-harboring. It has permitted us to highlight one of the consistent strategical requirements to better clarify the complexity of the AMR phenomenon which involves human, animal, and environmental health.

## 2. Materials and Methods

### 2.1. Samples: Listeria Monocytogenes Isolates

A total of 120 *Listeria monocytogenes* strains, belonging to our academic bacterial microbank (Food Inspection laboratories at Department of Veterinary Medicine, University of Teramo, Italy), were involved in this qualitative biomolecular study. Between 2008 and 2009, the above-mentioned strains were isolated from the seafood industry, and in more detail, from Atlantic salmon (*Salmo salar*) finfish species as fresh and processed products (salted and smoked food-processing) and from environmental swabs (collected from industrial surfaces, tools such as knives, and machines). The involved strains were successively identified through the biochemical method using the API^®^ LISTERIA kit (bioMérieux, Paris, France), and confirmed using PCR assays (amplifying specific sequences belonging to the 16S rRNA and listeriolysin O genes) [23]. Bacteria were then stored at −80 °C, as stock cultures, using Microbank vials (Biolife Italiana, Milan, Italy).

The analytical steps performed in the present scientific investigation started from *Listeria* isolates recovered using selective culture media performed in agreement with the International Microbiological Procedures: ISO 11290-1:2017 [24]. For each *Listeria* strain, a Microbank aliquot (1 glycerol plastic sphere) was introduced in a sterile plastic vial (with a final volume of 15 mL) containing 5 mL of Listeria Fraser broth (ThermoFisher Sceintitific^TM^, Milan, Italy) (as indicated by the international procedure: ISO 11290-1:2017). These ones were successively incubated at 37 °C for 24–36 h. During all microbiological analyses, ATCC 19,115 (*L. monocytogenes*) was used as a positive control for the performed screenings and was obtained from the American Type Culture Collection (Manassas, VA, USA).

### 2.2. Antimicrobial Susceptibility Tests (ASTs)

After the incubation period in the above-mentioned broth, *L. monocytogenes* strains were directly plated onto Mueller–Hinton agar (ThermoFisher Sceintitific^TM^, Milan, Italy) to perform the ASTs, in agreement with the disc diffusion Kirby–Bauer method (ISO 16782:2016). The breakpoints values were interpreted following the guidelines provided by the European Committee on Antimicrobial Susceptibility Testing (EUCAST) schemes [25].

Based on the last discovered resistance patterns, observed in many studies on *L. monocytogenes* isolates [16,26,27,28], sixteen antibiotic molecules used for listeriosis treatments among veterinary and human medicine were selected and tested. The panel included ampicillin (AMP 10 µg), ampicillin/sulbactam (10/10 µg), amoxycillin/clavulanic acid (AUG 10 µg), clindamycin (CLI 10 µg), chloramphenicol (C 30 µg), gentamycin (CN 10 µg), doxycycline (DXT 30 µg), erythromycin (E 15 µg), kanamycin (KAN 30 µg), linezolid (LNZ 30 µg), meropenem (MEM 10 µg), penicillin (P 10U), rifampicin (RD 5 µg), sulfamethoxazole and trimethoprim (SXT 23.27/1.25 µg), tetracycline (TET 30 µg), and vancomycin (VAN 30 µg). The inhibiting zones were precisely measured and strains resistant to at least three different antibiotic classes were permitted to classify the investigated strains as multidrug resistant (MDR), as previously reported by many authors [16,26].

In this paper, the applied *scientific workflow* began with the obtained ASTs results and continued through the PCR assays (set as uniplex or multiplex assays) for the detection of specific target genes encoding the phenotypical antibiotic resistances. In Section 2.5, gene determinants, amplified from both resistant and susceptible strains, were included.

### 2.3. DNA Extraction: PCR Assays

From the cultured broths, aliquots of 100 µL of each *Listeria* isolate were collected and used for the DNA extraction procedures, which were performed using High Pure PCR Template Preparation Kits (Roche^®^, Indianapolis, IN, USA). The nucleic acids were successively stored at −20 °C until the biomolecular assays, as described in the following subsections.

### 2.4. Serotyping and Virulence Factors

A number of 100 *Listeria* strains out of 120 were also screened through a multiplex PCR assay identifying specific target genes useful for serovars determination. A total of 20 strains were excluded because they were previously included and serotyped in another investigation [29]. The biomolecular reactions were performed in final volumes of 25 µL using a Green Master Mix Promega^®^ kit (Madison, WI, USA) with specific targeting primers for *lmo0737*, *lmo1118*, *ORF2819*, *ORF2110*, and *prs* genes detection, as previously performed by Doumith et al. [30]. Finally, an aliquot of 1 µL of DNA extracts was added in each reaction tube. Thermocycler settings and interpretations were executed as indicated by the scientific literature [30]. Successively, all the obtained amplicons were loaded on the agarose gel at 1.5% or 2% (depending on the amplicon sizes) and were compared to a specific DNA ladder (Genetics, Fast-Gene 100 bp DNA Marker) (NIPPON Genetics Europe GmbH, Düren, Germany).

*L. monocytogenes* strains were also tested for 10 virulence factors (*plc*A, *plc*B, *mpl*, *inl*A, *inl*B, *iap*, *rrn*, *hly*A, *act*A, and *porf*A) detection. Three different multiplex PCRs were performed for their detection [31,32]. The used primers permit to amplify target genes responsible for the encoding of any bacterial structures, the results of which are fundamental for the interaction between *L. monocytogenes* strains and host cells. All reactions were performed in a total volume of 50 µL, adding 2 µL of DNA using the above-mentioned kits. The thermocycler (Eppendorf^®^ Mastercycler Nexus X2 Thermal Cyclers, Hamburg, Germany) settings were set in agreement with references [31,32].

### 2.5. ARGs Detection

All phenotypical resistant and susceptible *L. monocytogenes* strains were involved in the ARGs biomolecular screenings. Depending on the used primers (reported in Table 1), multiplex and uniplex PCRs were performed to discover amplicons encoding resistance determinants against the commonly used antibiotic molecules in veterinary and human medicine (including CIA ones).

### 2.6. Statistical Analysis

Statistical analysis included a one-way variance ANOVA associated to a post hoc Tukey test (supported by XLSTAT 2014 software^®^ (Renmond, Washington, DC, USA)), which was used to identify significant differences regarding the antibiotic resistance patterns (phenotypic and ARGs) that were shown by the screened *L. monocytogenes* strains, as reported by Şanlıbaba et al. [46]. Results were considered significant if *p*-values were <0.05. For all calculated percentages, when applicable, we also determined the confidence intervals (CI: 95%).

## 3. Results

### 3.1. ASTs

Among *L. monocytogenes* isolates (n. 120), 63/120 or 52.50% (95% CI: 43.57–61.43%) strains were resistant to at least one screened antibiotic class. Concerning the tested antibiotic molecules, the most representative resistances were firstly observed against macrolides (chloramphenicol and erythromycin), followed by aminoglycosides (gentamycin and kanamycin), glycopeptide (vancomycin), oxazolidinone (linezolid), and tetracycline (tetracycline).

The resistant isolates majorly showed the following phenotypic patterns (focusing on resistance results/antibiotic molecules): erythromycin (35/120 or 29.16%; 95% CI: 21.03–37.29%), gentamycin (13/120 or 10.83%; 95% CI: 5.27–16.39%), kanamycin (15/120 or 12.50%; 95% CI: 6.58–18.42%), linezolid and tetracycline (22/120 or 18.33%; 95% CI: 11.41–25.25%), and vancomycin (18/120 or 15.00%; 95% CI: 8.62–21.38%).

A number of 25 out of 120 strains, which represented 20.83% (95% CI: 13.57–28.09%) among the total number and 39.68% (95% CI: 27.60–51.76%) regarding not-susceptible strains, showed resistance to three or more antibiotic classes, and for this reason were classified as multidrug resistant (MDR). In more detail, among them, 21/25 (84.0%; 95% CI: 69.63–98.37%) were discovered from smoked products, and the remaining 4/25 (16.0%; 95% CI: 1.63–30.37%) were identified as follows: half from knives and the other ones from machines surfaces. In detail, phenotypic resistance distributions related to the isolation sources are reported in the following Table 2.

Among strains isolated from fresh vs. smoked products, there was a significant difference regarding phenotypic resistance patterns, showing a *p*-value < 0.0001.

### 3.2. ARGs Detection

All screened *L. monocytogenes* isolates, including resistant and susceptible ones, were tested thorough biomolecular assays for ARGs detection. Phenotypic resistance patterns were genotypically confirmed, but numerous susceptible strains also harbored them. A detailed representation is illustrated by Figure 1.

In more detail, among the screened ARGs, tetracycline ones: *tet*C (53/120 strains 44.16%; 95% CI: 35.28–53.04%), *tet*D (70/120 strains 58.33%; 95% CI: 49.51–67.15%), *tet*K (62/120 strains 51.66%; 95% CI: 42.72–60.60%), *tet*L (98/120 strains 81.66%; 95% CI: 74.74–88.58%), and *tet*S (48/120 strains 40.0%; 95% CI: 31.24–48.76%) were majorly amplified compared with the other target genes from several samples. The above-mentioned ARGs were mainly discovered from processed specimens. The remaining screened genes, *tet*A, *tet*B, *tet*M(1), *tet*M(2), and *tet*O, were rarely amplified from *Listeria* strains.

This antibiotic class was followed by aminoglycosides *aad*A and *str*A (63/120 strains 52.50%; 95% CI: 43.57–61.43%), *aac*C2 (23/120 strains 19.16%; 95% CI: 12.12–26.20%), and *aph*A1 and *aph*A2 (31/120 strains 25.83%; 95% CI: 18.0–33.66%). Among macrolides, *cml*A1 (51/120 strains 42.50%; 95% CI: 33.65–51.35%), *cat*I (8/120 strains 6.67%; 95% CI: 2.21–11.13%), and *cat*II (17/120 strains 14.16%; 95% CI: 7.93–20.39%) were amplified. The oxazolidinones group amplified *cfr* (23/120 strains 19.16%; 95% CI: 12.12–26.20%), and *optr*A and *poxt*A (25/120 strains 20.83%; 95% CI: 13.57–28.09%) target genes.

The beta-lactam class, and its respective determinants, was low amplified; indeed, *bla*_Z_ was discovered from only one isolate (from the smoked salmon), and the other included ARGs (*amp*C and *bla*_TEM_) were low represented when compared with the other molecules (with special regard to the tetracyclines).

Concerning the vancomycin antibiotic genetic determinants *van*M was never amplified and *van*A, *van*B, *van*C1, *van*C2, *van*D, and *van*N were marginally amplified. Finally, the lincomycin class ARGs (*erm*A, *erm*B, *erm*C), represented by the clindamycin antibiotic molecule, were not discovered.

A general distribution, regarding the mainly detected determinants, is represented in the following heatmap in Figure 2.

### 3.3. Serovars and Virulence Factors

Following the interpretation schemes of Doumith et al. [30], multiplex PCR results for *L. monocytogenes* serovars showed different ones, which are illustrated in detail in the following Table 3.

Regarding the identified serovars, the 1/2a one was majorly discovered from all isolates representing 58.33% (95% CI: 49.51–67.15%), followed by 1/2b representing 15.83% (95% CI: 9.3–22.36%), and finally serovar 4d being 8.33% (95% CI: 3.39–13.27%).

The 10 virulence genes (*plc*A, *plc*B, *mpl*, *inl*A, *inl*B, *iap*, *rrn*, *hly*A, *act*A, *prf*A) were widely amplified from all *L. monocytogenes* isolates identified in the sampled sources. Any exceptions were represented by 6/10 (60%; 95% CI: 29.64–90.36%) strains, isolated from fresh salmon products, in which *pcl*B, and *mpl* target genes were not discovered. Only 1 out of 74 strains, isolated from processed products, did not show the following genetic determinants: *mpl*, *rrn*, *hyl*A, *act*A, and *prf*A.

### 3.4. Statistical Analysis

From a macroscopic perspective, in comparing phenotypical resistant and susceptible strains, we observed a statically different ARGs distribution with a *p*-value: <0.0001. In more detail, tet genetic determinants (*tet*C, *tet*D, *tet*L, and *tet*S) (see Figure 2) were mostly amplified. These ones presented a final *p*-value: <0.0001 if compared with the other detected ARGs encoding resistance against other screened antibiotic families, such as aminoglycosides (*aad*A, *str*A, *aph*A1, and *aph*A2), macrolides (*cml*A1 and *cat*II), and glycopeptides (*cfr*, *optr*A, *poxt*A). In the tetracycline antibiotic class, *tet*L was significantly (*p*-value: 0.02) amplified if compared to the other tet oligonucleotide determinants. Similar results (*p*-values: <0.0001) were observed in the other antibiotic families, such as aminoglycosides (*aad*A and *str*A), macrolides (*cml*A1), and oxazolidinones (*cfr*).

Based on the different isolation sources, ARGs were widely discovered from *L. monocytogenes* cultured from smoked salmon samples; indeed, it a significant difference was observed in these compared with environmental ones (machines, knives, and surfaces) presenting *p*-value: 0.0012 (see detailed ARGs distributions in Figure 2).

## 4. Discussion

In this investigation, based on culture-dependent analysis, biomolecular screenings were applied regarding ARGs detection from *L. monocytogenes* strains isolated in an integrated fish food supply. Based on a One-Health perspective, end-point PCR assays were performed including all *Listeria* isolates discovered from raw materials, salting and smoking processes (*Salmo salar* fillets), and involving environmental swabs. The adopted molecular approach was used in order to focus scientific attention on comparisons of the ARG distributions between the amplified ones from 2008-aged *L. monocytogenes* strains with the recent genetic harboring from the literature. Starting from this perspective, an evaluation of potential risks for potential horizontal transmission between food matrices and consumer microbiota has relevant repercussions at the industry level. Therefore, this investigation has focused its attention on an important foodborne pathogen, *L. monocytogenes*, and due to its microbiological importance, as foodborne *noxa*, it was included in the so-called food safety criteria by the European Legislator (in accordance with the EU Reg. No. 2073/2005).

The related risk assessment, EU Reg. No. 765/2008 [47], which is normally applied in food-producing industries, will be involved in and responsible for the further repercussions and diffusion of ARGs. This screening of the AMR phenomenon has potential repercussions for the final consumers due to the ingestion of animal origin foodstuffs [48]. This last consideration is based on the well-known correlation regarding possible horizontal transmission of mobile resistant determinants between food matrices’ microbiota with the human intestinal one [49]. For these reasons, in the present study, antibiograms and biomolecular assays also involved both veterinary (majorly used in finfish aquaculture) and the CIA molecules, whose administration is highly recommended for humans only [12].

Concerning the obtained phenotypic resistance patterns, erythromycin presented the highest resistance percentages with 35/120 or 29.16% and 95% CI: 21.03–37.29%, if compared with the other screened molecules: gentamycin (13/120 or 10.83%; 95% CI: 5.27–16.39%), kanamycin (15/120 or 12.50%; 95% CI: 6.58–18.42%), linezolid and tetracycline (22/120 or 18.33%; 95% CI: 11.41–25.25%), and vancomycin (18/120 or 15.00%; 95% CI: 8.62–21.38%).

More specifically, a parallel resistance was observed against erythromycin, gentamycin, and tetracycline in 18/120 (15.0%; 95% CI: 8.62–21.38%) *Listeria* strains, representing 64.0% (95% CI: 45.19–82.81%) of MDR strains. These results were in line with previous and recent studies that isolated *L. monocytogenes* through culture-dependent methods, from ready-to-eat fish products [15,50,51] and fresh fish fillet samples [16,22,52]; and a wide resistance against the linezolid molecule was observed [53]. The above-mentioned antibiotic molecules, i.e., erythromycin, gentamycin, vancomycin, linezolid, etc., are normally used for listeriosis treatments in human infection cases. For this reason, it is consequently important to preserve their efficacy. The scientific community has widely demonstrated that genetic mobile elements (i.e., plasmids) are involved in the enforcement of this pathway [54]. An historical scientific representation from 1950 to 2021 highlighted the notable variations of phenotypic resistance patterns, observing decreasing resistance against the first antibiotic molecules (i.e., penicillin) and the consistent increase against aminoglycosides, macrolides, beta-lactams, etc. [55]. The critical microbiological and molecular situations, denounced by Andriyanov et al. [55], have been also enforced by the obtained data belonging to the present scientific investigation. Excluding possible intrinsic resistances, evolution has selected resistant *L. monocytogenes* strains, through a “*positive pressure*”, which phenotypically express for modified efflux pumps which are not receptive to the biochemical interaction between chemical ligands (antibiotics) and bacterial parietal receptors (as trans-membrane structures). This pathway provides substantial survival structures, facilitating biofilm formation and the AMR spreading in food industries [56].

In this study, it was observed that all resistant *Listeria* strains were genotypically positive for respective ARGs detection; furthermore, the discovered biomolecular scenario provided fascinating data highlighting a consistent ARGs distribution among susceptible strains, too. Indeed, a statistically significant difference was reported, if the discovered gene determinants’ amounts, amplified from phenotypically resistant and susceptible isolates, were compared (*p*-value < 0.0001). A detailed representation about harboring strains’ distributions is illustrated by Figure 1 (Results section). These findings represent a clear confirmation regarding the so-called scientific theory of “*ARGs different expression*” in which genes’ phenotypical expression depends on several environmental stressors (i.e., massive antibiotic administration). These are consequences of “*genetic plasticity*”, which are adaptive responses to mutational alterations [57]. Comparing all ARGs distributions between food matrices and environmental Listeria isolates, we observed significant differences (*p*-value < 0.002), in agreement with Chen et al. [58], Jamali et al. [59], and Wiśniewski et al. [60]. In more detail, the *tet* genes *tet*L, *tet*D, *tet*C, and *tet*S were largely amplified. Indeed, we observed a further statistically significant difference between smoked and environmental strains (*p*-value < 0.0001). These findings were partially in agreement with *tet*C and *tet*S distributions, while *tet*A and *tet*M were also widely amplified by Jamali et al. [59] and Parra-Flores et al. [61], in line with contemporary studies performed in the last decade (years 2007–2009) [15,16]. These differences show as the consistent plasticity of the evolutive pressures related to the improper administration of these molecules.

In the present study, similar trends were also discovered for aminoglycosides (*aad*A, *str*A, *aph*A1, and *aph*A2), macrolides (*cml*A1 and *cat*II), and glycopeptides (*cfr*, *optr*A, *poxt*A) ARGs (with special regard to the susceptible ones), as illustrated in the respective Figure 1 and Figure 2. These data were in line with those described by many studies [15,16,62] for aminoglycosides, macrolides [57], and, only recently, glycopeptides (linezolid) [55]. More in detail, linezolid ARGs-harboring was never observed in 2008; however, in recent studies from 2018–2022, scientists mainly amplified *cfr* and *poxt*A genes [55,62]. The possibility of multiple resistances has been scientifically demonstrated by the identification of gene cassettes which are involved as ARGs drivers, i.e., *tet*S, *aad*A, and *cfr* [63] and *tet* genes coupled with chloramphenicol ones [20]. Following the ARGs detection discussed above, many determinants, belonging to the CIA molecules, were also amplified. *Van*D (glycopeptide class), *cfr*, *optr*A, and *poxt*A (oxazolidinone) were majorly discovered from smoked and environmental samples (surface swabs) with the following percentage values: 6.67% (95% CI: 2.21–11.13%) and 16.67% (95% CI: 10.0–23.34%), respectively. These genetic determinants were not observed in the 2008 studies but resulted in agreement with ones reported by another recent investigation [64] that interpreted these patterns due to the environmental pollution and to the consistent harboring in commensal bacteria as “*environmental reservoirs*”.

Among the screened genes included in this study, *erm*A, *erm*B, and *erm*C (belonging to the lincomycin antibiotic class) were not detected from the discovered serotypes 1/2a, 1/2c, and 4d. These findings confirmed the scientific hypothesis proposed by a previous investigation [65], which found further confirmation with recent ones [66,67] that correlated *L. monocytogenes* serotypes 1/2a and 1/2c to the frequent harboring of tetracycline ARGs that lacked erm ones. The whole genome sequencing produced an evolutionary explanation identifying *erm* as a gene’s loss during bacterial multiplication cycles due to the decreased lincomycin usage for listeriosis treatments [68].

For this reason, *erm* genes absence could be explained by two reasonable scientific hypotheses: the first one could be related to the so-called *gene loss phenomenon*, and the second one could be explained by probable oligonucleotide mutations.

Numerous studies have further investigated the possible correlation between virulence genes and ARGs detection. In this investigation, most *Listeria* isolates harbored different virulence factors (i.e., *plc*A, *plc*B, *mpl*, *inl*A, *inl*B, *iap*, *rrn*, *hly*A, *act*A, *prf*A) that also resulted in being resistant to at least one antibiotic class. These findings, in agreement with a previous research study [58] compared with recent ones [49,50,59,62] among different animal origin foodstuffs, have identified their (virulence genes) simultaneous detections with ARGs from MDR *L. monocytogenes* strains. The involved antibiotic classes were aminoglycosides, beta-lactams, macrolides, tetracyclines, etc. Indeed, as discussed above, biomolecular technologies as the proteomic characterization (through the usage of liquid chromatography) coupled with whole genome sequencing have demonstrated and enforced the growing statistical correlations between virulence genes and ARGs detection [69]. Following this scientific scheme, phylogenetic analysis has highlighted how these oligonucleotide determinants, isolated from different food matrices, have strict correlations with environmental ones, based on their evolutionary aspects [70]. As considered above, this investigation, conducted in an “*industrial environment*”, aimed to provide interesting preliminary data about ARGs circulation amplified from *L. monocytogenes* isolates. The obtained results of this retrospective study from *L. monocytogenes* strains (15 years ago), permitted us to enforce the consideration of the incisive evolution and pleomorphism which characterize the difficult side of this scientific topic. This fascinating genotypic scenario has allowed us to discover similar ARGs distributions comparing 2008-aged *Listeria* strains to the recent ones. The importance of research and molecular investigations focuses further attention on the wide evolutionary ARGs trades which have increased over the years, as also confirmed by many previous or recent studies [15,20,55]. Due to the complexity of the AMR phenomenon, further studies will be performed in order to satisfy and remove any gaps in knowledge. Currently, integrated approaches involving salmon food matrices and industrial environmental samples has been not yet conducted. For this reason, this study represents the basis on which will be designed other scientific articles concerning the Atlantic salmon (*Salmo salar*) industry and the topic of antibiotic resistance research.

## 5. Conclusions

*L. monocytogenes* represents one the most dangerous and zoonotic microorganisms for food industries. Moreover, the AMR phenomenon has involved both pathogen and commensal strains, and ARGs environmental diffusion has become one of the crucial challenges for the near future. Due to “*genetic plasticity*”, genetic determinants will be characterized by numerous mutations, inducing a growing necessity to improve innovative technologies to act as advanced diagnostic systems (i.e., whole genome sequencing). The obtained results showed, in agreement with previous 2008-aged and contemporary studies, a wide circulation of tet genes. However, in contrast to *tet*K, *tet*L, *tet*M(1), and *tet*M(2), which were also amplified 15 years ago, new modified oligonucleotide sequences *tet*A, *tet*B, *tet*C, *tet*D, and *tet*S were also discovered. This scenario can be justified by the wide tetracycline class administration in both human and veterinary medicine. This has been further enriched by the alarming amplification of ARGs encoding resistance against the CIA molecules. Indeed, in this study, *cfr*, *optr*A (linezolid), *cml*A1, *cat*II (chloramphenicol and erythromycin), *van*D, and *van*N (vancomycin) were discovered. These last findings were not observed or described by other authors in the scientific literature 15 years ago. These scientific gaps can be explained by focusing on two critical aspects. The first one is represented by a consistent selective pressure which was induced by massive or improper antimicrobial administrations (in both human and veterinary medicine) without preserving molecules of crucial importance for the human species. These micro-environmental conditions have determined the survival of multidrug or pan-resistant bacteria. The second one is strictly related to the low sensibility of molecular diagnostic methods, and the low interests for the AMR scientific topic especially for food-isolated bacteria. The aim of this study was to focus on the risk associated with the ARGs biochemical trades between commensal and pathogenic strains. The real concern is represented by the bacteria which harbor many genetic determinants acting as vectors or reservoirs for other strains. The aim is the reduction of a consistent selective pressure, and secondly, the improvement of valid substitutions (i.e., vaccination programs, innovative and sustainable farming systems in the aquaculture sector, probiotic administrations, etc.) which will be suitable solutions able to reduce the AMR global impact. However, conversely to the European and North American estimations, concerning the antibiotic consumptions expressed as tons, the FAO estimated that the so-called BRICS nations (Brazil, Russia, India, China, and South Africa) will experience an increase with special regard to the mammalian and aquaculture zootechnic sectors. A common and international solution has become strictly necessary to preserve the efficacy of certain molecules. Therefore, basing on a One Health approach, the human species has a crucial responsibility to preserve animal and environmental health. 

## Figures and Tables

**Figure 1 microorganisms-11-01509-f001:**
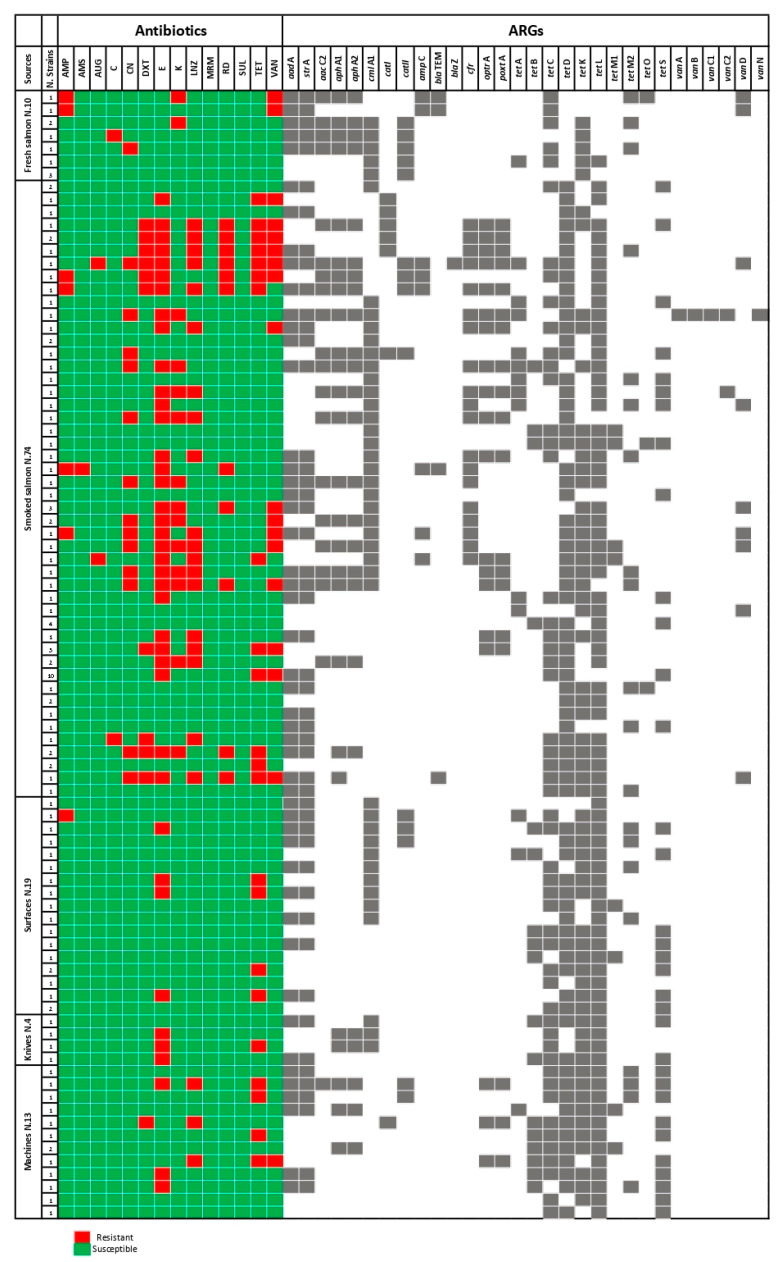
Phenotypic and genotypic antibiotic susceptibility patterns discovered in *L. monocytogenes* strains isolated from the Atlantic salmon (Salmo salar) fish industry. Red squares indicate resistant bacteria and the green ones represent susceptible strains. The second part (on the right) illustrates ARGs distribution among both phenotypically and genotypically resistant *L. monocytogenes* isolates. AMP: ampicillin; AMS: ampicillin–sulbactam; AUG: amoxicillin–clavulanic acid; CLI: clindamycin; C: chloramphenicol; CN: gentamycin; DXT: doxycycline; E: erythromycin; KAN: kanamycin; LNZ: linezolid; TET: tetracycline; VAN: vancomycin.

**Figure 2 microorganisms-11-01509-f002:**
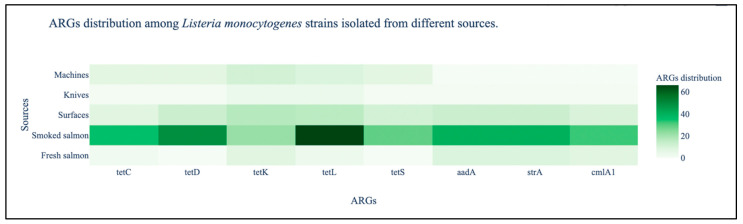
Heatmap based on ARGs distribution among *L. monocytogenes* per food matrices or environmental specimens.

**Table 1 microorganisms-11-01509-t001:** Target genes, oligonucleotide sequences, and references for ARGs detection.

Antibiotic Class	Genes	Nucleotide Sequences	Amplicon Sizes	M/U	References
Aminoglycosides (CN; KAN)	*aad*A	F: GTGGATGGCGGCCTGAAGCC R: AATGCCCAGTCGGCAGCG	525 bp	M	[33]
	*str*A	F: CTTGGTGATAACGGCAATTC R: CCAATCGCAGATAGAAGGC	348 bp		
	*aac*C2	F: CGGAAGGCAATAACGGAG R: TCGAACAGGTAGCACTGAG	428 bp	M	[34]
	*aph*A1	F: ATGGGCTCGCGATAATGTC R: CTCACCGAGGCAGTTCCAT	600 bp		
	*aph*A2	F: GAACAAGATGGATTGCACGC R: GCTCTTCAGCAATATCACGG	510 bp		
Lincomycin (CLI)	*erm*A	F: GTTCAAGAACAATCAATACAGGAG R: GGATCAGGAAAAGGACATTTTAC	421 bp	M	[35]
	*erm*C	F: GCTAATATTGTTTAAATCGTCAATTCC R: GGATCAGGAAAAGGACATTTTAC	572 bp		
	*erm*B	F: GAAAAGGTACTCAACCAAATA R: AGTAACGGTACTTAAATTGTTTAC	639 bp	U	[36]
Macrolides (C; E)	*cml*A1	F: CACCAATCATGACCAAG R: GGCATCACTCGGCATGGACATG	115 bp	U	[37]
	*catI*	F: AGTTGCTCAATGTACCTATAACC R: TTGTAATTCATTAAGCATTCTGCC	320 bp	M	[33]
	*catII*	F: ACACTTTGCCCTTTATCGTC R: TGAAAGCCATCACATACTGC	543 bp		
Beta-lactams (AMP; AMS; AUG)	*amp*C	F: TTCTATCAAMACTGGCARCC R: CCYTTTTATGTACCCAYGA	550 bp	U	[32]
	*bla* _TEM_	F: TTTCGTGTCGCCCTTATTCC R: CCGGCTCCAGATTTATCAGC	690 bp	U	[37]
	*bla*Z	F: ACTTCAACACCTGCTGCTTTC R: TGACCACTTTTATCAGCAACC	490 bp	U	[38]
Oxazolidinone (LNZ)	*cfr*	F: TGAAGTATAAAGCAGGTTGGGAGTC R: AACCATATAATTGACCACAAGCAGC	746 bp	M	[39]
	*optr*A	F: TACTTGATGAACCTACTAACCA R: CCTTGAACTACTGATTCTCGG	422 bp		
	*poxt*A	F: AAAGCTACCCATAAAATATC R: TCATCAAGCTGTTCGAGTTC	533 bp		
Tetracycline (DXT; TET)	*tet*A	F: GCTACATCCTGCTTGCCTTC R: CATAGATCGCCGTGAAGAGG	210 bp	M	[40]
	*tet*B	F: TTGGTTAGGGGCAAGTTTTG R: GTAATGGGCCAATAACACCG	659 bp		
	*tet*C	F: CTTGAGAGCCTTCAACCCAG R: ATGGTCGTCATCTACCTGCC	418 bp		
	*tet*D	F: AAA CCA TTA CGG CAT TCT GC R: GACCGGATACACCATCCATC	787 bp		
	*tet*S	F: CATAGACAAGCCGTTGACC R: ATGTTTTTGGAACGCCAGAG	667 bp		
	*tet*M(1)	F: GGTACTTGAAAAGAACGGGAG R: TTCACCTTAGTATTTTTCCACTG	630 bp	M	[41]
	*tet*M(2)	F: GGTACTTGAAAAGAACGGGAG R: ATACGAGTTTGTGCTTGTACGCC	740 bp		
	*tet*K	F: TATTTTGGCTTTGTATTCTTTCAT R: GCTATACCTGTTCCCTCTGATAA	1159 bp	M	[42]
	*tet*L	F: ATAAATTGTTTCGGGTCGGTAAT R: AACCAGCCAACTAATGACAATGAT	1077 bp		
	*tet*O	F: AACTTAGGCATTCTGGCTCAC R: TCCCACTGTTCCATATCGTCA	519 bp	U	[43,44]
Glycopeptide (VAN)	*van*A	F: GCAAGTCAGGTGAAGATGGA R: GCTAATACGATCAAGCGGTC	171 bp	M	[45]
	*van*B	F: GATGTGTCGGTAAAATCCGC R: CCACTTCGCCGACAATCAAA	271 bp		
	*van*C1	F: GTATCAAGGAAACCTCGCGA R: CGTAGGATAACCCGACTTCC A	836 bp		
	*van*C2	F: GCAAACGTTGGTACCTGATG R: GGTGATTTTGGCGCTGATCA	523 bp		
	*van*D	F: TGGAATCACAAAATCCGGCG R: TWCCCGCATTTTTCACAACS	311 bp		
	*van*M	F: GGCAGAGATTGCCAACAACA R: AGGTAAACGAATCTGCCGCT	425 bp		
	*van*N	F: CCTCAAATCAGCAGCTAGTG R: GCTCCTGATAAGTGATACCC	941 bp		

M: multiplex PCR; U: uniplex PCR. AMP: ampicillin; AMS: ampicillin–sulbactam; AUG: amoxicillin–clavulanic acid; CLI: clindamycin; C: chloramphenicol; CN: gentamycin; DXT: doxycycline; E: erythromycin; KAN: kanamycin; LNZ: linezolid; TET: tetracycline; VAN: vancomycin.

**Table 2 microorganisms-11-01509-t002:** ASTs: phenotypic pattern results.

Sources		N. Strains	Resistant Strains	Antibiotic Molecules
Food matrices 84/120	Fresh salmon	10/84	3/10	K
			2/10	AMP, VAN
			1/10	C
			1/10	CN
	Smoked salmon	74/84	30/74	E
			19/74	LNZ
			15/74	VAN
			13/74	CN, TET
			12/74	K
			11/74	RD
			10/74	DXT
			0/74	CLI
			4/74	AMP
			2/74	AUG
			1/74	AMS, C
Environment 36/120	Surfaces	19/36	4/19	E, TET
			1/19	AMP
	Knives	4/36	3/4	E
			1/4	TET
	Machines	13/36	4/13	TET
			3/13	E, LNZ
			1/13	DXT, VAN

AMP: ampicillin; AMS: ampicillin–sulbactam; AUG: amoxicillin–clavulanic acid; CLI: clindamycin; C: chloramphenicol; CN: gentamycin; DXT: doxycycline; E: erythromycin; KAN: kanamycin; LNZ: linezolid; TET: tetracycline; VAN: vancomycin.

**Table 3 microorganisms-11-01509-t003:** *L. monocytogenes* strains identified in this study.

Isolates	N. *L. monocytogenes*	Sources	Serovars
n. 120 *L. monocytogenes* strains	n. 10	Fresh salmon	n. 5 serovars 4d
			n. 5 serovars 1/2c
	n. 74	Smoked salmon	n. 39 serovars 1/2a
			n. 19 serovars 1/2b
			n. 6 serovars 4e
			n. 5 serovars 4b
			n. 5 serovars 4d
	n. 19	Surfaces	n. 18 serovars 1/2a
			n. 1 serovar 1/2c
	n. 4	Knives	n. 4 serovars 1/2a
	n. 13	Machines	n. 9 serovars 1/2a
			n. 3 serovars 1/2c
			n. 1 serovar 4b

## Data Availability

Data are contained within the article.

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
