# Peer review of "ARGs Detection in *Listeria Monocytogenes* Strains Isolated from the Atlantic Salmon (*Salmo salar*) Food Industry: A Retrospective Study"

_microorganisms, 2023, doi:10.3390/microorganisms11061509_

Round 1

Reviewer 1 Report

Dear Authors

Listeria monocytogenes  and antibiotic resistance poses a significant threat to public health due to its ability to cause severe foodborne illnesses, particularly in vulnerable populations.

The study was designed as a retrospective study, which aimed to investigate the trend of antibiotic resistance genes (ARGs) circulation in L. monocytogenes isolates   identified (in the last fifteen years) from Atlantic salmon (Salmo salar) fresh and smoked fillets and environmental samples.

Concerning ARGs   circulation, tetracyclines (tetC, tetD, tetK, tetL, tetS), aminoglycosides (aadA, strA, aacC2, aphA1, aphA2), macrolides (cmlA1, catI, catII), and oxazolidinones (cfr, optrA, poxtA) gene determinants were majorly amplified.

The study from a scientific point of view seems to be well done and presents good results, from where to derive valid conclusions. The paper is original and contains certain novelties.

The summary sufficiently informs about the content of the paper. But, abstract can be improved if include information about 95% confidence intervals.

Introduction summarizes relevant research to provide context and clearly state the problem.  The topics are well developed and confronted to other publications.

The research methods used ensure the reliability of the obtained results. They can be improved if include information about sample size calculation.

The interpretation of the results is correct.

Author Response

Dear Reviewer 1,

We are really pleasured that our article entitled “ARGs detection in Listeria monocytogenes strains isolated from an Atlantic salmon (Salmo salar) food industry: A retrospective study” has been considered for publication in the MICROORGANISMS Journal in the Special Issue “n Update on Listeria monocytogenes 2.0”.

All observations resulted precious for its scientific quality improvement.

In the following paragraphs we reported all changes.

Reviewer 1 suggestions:

Listeria monocytogenes and antibiotic resistance poses a significant threat to public health due to its ability to cause severe foodborne illnesses, particularly in vulnerable populations.

The study was designed as a retrospective study, which aimed to investigate the trend of antibiotic resistance genes (ARGs) circulation in L. monocytogenes isolates identified (in the last fifteen years) from Atlantic salmon (Salmo salar) fresh and smoked fillets and environmental samples.

Concerning ARGs   circulation, tetracyclines (tetC, tetD, tetK, tetL, tetS), aminoglycosides (aadA, strA, aacC2, aphA1, aphA2), macrolides (cmlA1, catI, catII), and oxazolidinones (cfr, optrA, poxtA) gene determinants were majorly amplified.

The study from a scientific point of view seems to be well done and presents good results, from where to derive valid conclusions. The paper is original and contains certain novelties.

The summary sufficiently informs about the content of the paper. But, abstract can be improved if include information about 95% confidence intervals.

Authors (lines 15-17): Abstract revised.

Introduction summarizes relevant research to provide context and clearly state the problem.  The topics are well developed and confronted to other publications.

The research methods used ensure the reliability of the obtained results. They can be improved if include information about sample size calculation.

Authors: The studied Listeria strains are part of our academic microbiological bank. We purposed to present and show graphically our results using tables (Table 2) and a specific figure (Figure 1). We hope that we provide a well-structured explanation. Thank You again for collaboration and for precious suggestions.

The interpretation of the results is correct.

We confirm that neither the manuscript nor any parts of its content are currently under consideration or published in another journal.

All authors have approved the manuscript and agree with its submission to the MICROORGANISMS Journal.

We appreciate the possibility to publish our paper and believe that our manuscript will be of interest to You and to the readers of Your journal.

Thank You for Your time and attention.

Best regards,

Gianluigi Ferri

Doctor in Veterinary Medicine (D.V.M.)

Ph.D. in Food Inspection

Department of Veterinary Medicine; University of Teramo, Italy.

Reviewer 2 Report

M.S. Microorganisms-2441266

The paper entitled “ARGs detection in Listeria monocytogenes strains isolated from an Atlantic salmon (Salmo salar) food industry: A retrospective study”, is a retrospective study, in which the Authors investigate investigate the trend of antibiotic resistance genes (ARGs) circulation in L. monocytogenes isolates identified (in the last fifteen years) from Atlantic salmon (Salmo salar) fresh and smoked fillets and environmental samples. From the biomolecular assays performed on 120 L. monocytogenes strains collected in recent years were compared with contemporary scientific literature; 52.50% were found to be resistant to at least one class of antibiotics and 20.83% were classified as multi-resistant. As far as the circulation of ARGs is concerned, the gene determinants of tetracyclines, aminoglycoside macrolides and oxazolidinones were most amplified. This study shows a consistent circulation of ARGs from fresh and processed fish products and environmental samples discovering resistance to so-called 'critically important antimicrobials' since 2007. The data obtained on the circulation of ARGs show a steady increase in their prevalence compared to similar contemporary surveys. This scenario emerges as the result of decades of improper administration of antimicrobials in human and veterinary medicine.

The study is interesting, but needs a major revision, especially in section 4 (discussion) and section 5 (conclusions).

Below my consideration line by line:

Line 35: “food safety criteria, as reported in the EU Reg. No. 2073/2005”. I suggest that the authors add the EU Reg. No. 2073/2005” in the references.

Line 52: tetracyclines [14,15,16]. Insert the references following the MDPI Style Guide. 

Line 87-88: A total of 120 Listeria monocytogenes strains, belonging to our Academic bacterial microbank, were involved in this qualitative bio-molecular study. Specify the laboratory(s): e.g. University laboratory, or .... others.

Line 114-115: observed in many studies from L. monocytogenes isolates [15,25,26,27],. Check L. monocytogenes and rewrite properly the references following the MDPI Style Guide. 

Line 127-128: For the ARGs detection and their relative circulation purpose, molecular screenings also included the susceptible ones (as described in the subsection 2.5. ARGs Detection). Rewrite the sentence clearly.

Line 164: in table 1, check “aphA2“.

Line 170: “Statistical analysis included the one-way variance ANOVA associated to Tukey test”. I suggest to Authors checking this form: 'associated to post-hoc Tukey test”.

Line 186: (35/120 or 29.16%- 95%. Check “%- 9”.

Line 246: Heatmap based on ARGs distribution among L. monocytogenes per food matrices. Rewrite in italics “L. monocytogenes”.

Line 263: “3.4. Statistical analysis”. “3.4. Statistical analysis” must be included in the materials and methods.

Line 293: “The related risk-assessment (EU Reg. No. 765/2008) which is normally applied in the …”.  I suggest that the authors add the “EU Reg. No. 765/2008” in the references.

Line: 278-471: “Discussion” and “Conclusions” are not well delineated. “Conclusions” should be short, concise and clear and the inclusion of references is not recommended.

I suggest to Authors that the “Discussion” and “Conclusions” should be rewrite and better organised.

Line 408: “L. monocytogenes”, rewrite in italics.

Minor editing of English language required.

Author Response

Dear Reviewer 2,

We are really pleasured that our article entitled “ARGs detection in Listeria monocytogenes strains isolated from an Atlantic salmon (Salmo salar) food industry: A retrospective study” has been considered for publication in the MICROORGANISMS Journal in the Special Issue “An Update on Listeria monocytogenes 2.0”.

All observations resulted precious for its scientific quality improvement.

In the following paragraphs we reported all changes.

Reviewer 2 suggestions:

The paper entitled “ARGs detection in Listeria monocytogenes strains isolated from an Atlantic salmon (Salmo salar) food industry: A retrospective study”, is a retrospective study, in which the Authors investigate the trend of antibiotic resistance genes (ARGs) circulation in L. monocytogenes isolates identified (in the last fifteen years) from Atlantic salmon (Salmo salar) fresh and smoked fillets and environmental samples. From the biomolecular assays performed on 120 L. monocytogenes strains collected in recent years were compared with contemporary scientific literature; 52.50% were found to be resistant to at least one class of antibiotics and 20.83% were classified as multi-resistant. As far as the circulation of ARGs is concerned, the gene determinants of tetracyclines, aminoglycoside macrolides and oxazolidinones were most amplified. This study shows a consistent circulation of ARGs from fresh and processed fish products and environmental samples discovering resistance to so-called 'critically important antimicrobials' since 2007. The data obtained on the circulation of ARGs show a steady increase in their prevalence compared to similar contemporary surveys. This scenario emerges as the result of decades of improper administration of antimicrobials in human and veterinary medicine.

The study is interesting, but needs a major revision, especially in section 4 (discussion) and section 5 (conclusions).

Below my consideration line by line:

Line 35: “food safety criteria, as reported in the EU Reg. No. 2073/2005”. I suggest that the authors add the EU Reg. No. 2073/2005” in the references.

Authors (line 35 and 479-480): Added references.

Line 52: tetracyclines [14,15,16]. Insert the references following the MDPI Style Guide.

Authors (line 73): Modified.

Line 87-88: A total of 120 Listeria monocytogenes strains, belonging to our Academic bacterial microbank, were involved in this qualitative bio-molecular study. Specify the laboratory(s): e.g. University laboratory, or .... others.

Authors (lines 90-91): Information added.

Line 114-115: observed in many studies from L. monocytogenes isolates [15,25,26,27],. Check L. monocytogenes and rewrite properly the references following the MDPI Style Guide.

Authors (lines 134-135): Modified.

Line 127-128: For the ARGs detection and their relative circulation purpose, molecular screenings also included the susceptible ones (as described in the subsection 2.5. ARGs Detection). Rewrite the sentence clearly.

Authors (lines 147-149): Sentence modified.

Line 164: in table 1, check “aphA2“.

Authors (line 222): “aphA2“ modified as “aphA2”.

Line 170: “Statistical analysis included the one-way variance ANOVA associated to Tukey test”. I suggest to Authors checking this form: 'associated to post-hoc Tukey test”.

Authors (lines 244-245): Modified.

Line 186: (35/120 or 29.16%- 95%. Check “%- 9”.

Authors (line 261): Modified.

Line 246: Heatmap based on ARGs distribution among L. monocytogenes per food matrices. Rewrite in italics “L. monocytogenes”.

Authors (line 327): Modified.

Line 263: “3.4. Statistical analysis”. “3.4. Statistical analysis” must be included in the materials and methods.

Authors: The “3.4 Statistical analysis” shows the obtained results from the statistical analysis which has been described in the Material and Methods section. In this part, we also provided the explanation of the applied statistical tests. For these reasons, it is our opinion that this section should be stable in the Results section.

Line 293: “The related risk-assessment (EU Reg. No. 765/2008) which is normally applied in the …”.  I suggest that the authors add the “EU Reg. No. 765/2008” in the references.

Authors (lines 375 and 783-785): Reference added.

Line: 278-471: “Discussion” and “Conclusions” are not well delineated. “Conclusions” should be short, concise and clear and the inclusion of references is not recommended.

I suggest to Authors that the “Discussion” and “Conclusions” should be rewrite and better organised.

Authors: These sections were modified following the provided suggestions.

Line 408: “L. monocytogenes”, rewrite in italics.

Authors (line 582): Modified.

We confirm that neither the manuscript nor any parts of its content are currently under consideration or published in another journal.

All authors have approved the manuscript and agree with its submission to the MICROORGANISMS Journal.

We appreciate the possibility to publish our paper and believe that our manuscript will be of interest to You and to the readers of Your journal.

Thank You for Your time and attention.

Best regards,

Gianluigi Ferri

Doctor in Veterinary Medicine (D.V.M.)

Ph.D. in Food Inspection

Department of Veterinary Medicine; University of Teramo, Italy.

Reviewer 3 Report

In the current study, Gianluigi Ferri et al. isolated 120 strains of Listeria monocytogenes and analyzed the association between their resistance to the antibiotics and the existence of antibiotic resistance genes. Overall, this is a very interesting and well-written paper. I only have few minor comments.

1) As can been seen from figure 1, it seems the resistance to several antibiotics such as E, LNZ, and TET widely exists in the tested strains. Is there any possible underlying reasons. The authors may discuss it in the manuscript.

2) The Listeria monocytogenes were collected from one company. Therefore, is there any possibility of cross-contamination between specimen? Besides, is there any sterilization protocols adopted in this factory which may result the bias in the antibiotic resistance?

3) The authors indicated that the susceptible strains also harbor the ARGs. It seems in the current study, the authors only detected the existence of ARGs. I would be curious that is there any differences in the copy numbers of ARGs among resistant and susceptible strains?

Author Response

Dear Reviewer 3,

We are really pleasured that our article entitled “ARGs detection in Listeria monocytogenes strains isolated from an Atlantic salmon (Salmo salar) food industry: A retrospective study” has been considered for publication in the MICROORGANISMS Journal in the Special Issue “n Update on Listeria monocytogenes 2.0”.

All observations resulted precious for its scientific quality improvement.

In the following paragraphs we reported all changes.

Reviewer 3 suggestions:

In the current study, Gianluigi Ferri et al. isolated 120 strains of Listeria monocytogenes and analyzed the association between their resistance to the antibiotics and the existence of antibiotic resistance genes. Overall, this is a very interesting and well-written paper. I only have few minor comments.

  • As can been seen from figure 1, it seems the resistance to several antibiotics such as E, LNZ, and TET widely exists in the tested strains. Is there any possible underlying reasons. The authors may discuss it in the manuscript.

Authors (lines 405-407): This part was implemented.

2) The Listeria monocytogenes were collected from one company. Therefore, is there any possibility of cross-contamination between specimen? Besides, is there any sterilization protocols adopted in this factory which may result the bias in the antibiotic resistance?

Authors: Cross-contaminations should not be excluded as possible infection sources. The screened companied was certified for international standards (ISO) and for this reason it had adapted protocols for sterilization.

3) The authors indicated that the susceptible strains also harbor the ARGs. It seems in the current study, the authors only detected the existence of ARGs. I would be curious that is there any differences in the copy numbers of ARGs among resistant and susceptible strains?

Authors: ARGs were mostly detected from resistant strains if compared with the susceptible ones.

We confirm that neither the manuscript nor any parts of its content are currently under consideration or published in another journal.

All authors have approved the manuscript and agree with its submission to the MICROORGANISMS Journal.

We appreciate the possibility to publish our paper and believe that our manuscript will be of interest to You and to the readers of Your journal.

Thank You for Your time and attention.

Best regards,

Gianluigi Ferri

Doctor in Veterinary Medicine (D.V.M.)

Ph.D. in Food Inspection

Department of Veterinary Medicine; University of Teramo, Italy.

Round 2

Reviewer 2 Report

The paper has been improved, congratulations to the Authors.